# Dynamics-Guided Diffusion Model
# for Sensor-less Robot Manipulator Design

**Xiaomeng Xu**[1]    **Huy Ha**[2]    **Shuran Song**[1]
[1]Stanford University    [2]Columbia University

https://dgdmcorl.github.io

**Abstract:** We present Dynamics-Guided Diffusion Model (DGDM), a data-driven framework for generating task-specific manipulator designs without task-specific training. Given object shapes and task specifications, DGDM generates sensor-less manipulator designs that can blindly manipulate objects towards desired motions and poses using an open-loop parallel motion. This framework 1) flexibly represents manipulation tasks as interaction profiles, 2) represents the design space using a geometric diffusion model, and 3) efficiently searches this design space using the gradients provided by a dynamics network trained without any task information. We evaluate DGDM on various manipulation tasks ranging from shifting/rotating objects to converging objects to a specific pose. Our generated designs outperform optimization-based and unguided diffusion baselines relatively by 31.5% and 45.3% on average success rate. With the ability to generate a new design within 0.8s, DGDM facilitates rapid design iteration and enhances the adoption of data-driven approaches for robot mechanism design. Qualitative results are best viewed on our project website https://dgdmcorl.github.io.

**Keywords:** manipulator design, hardware optimization, diffusion model

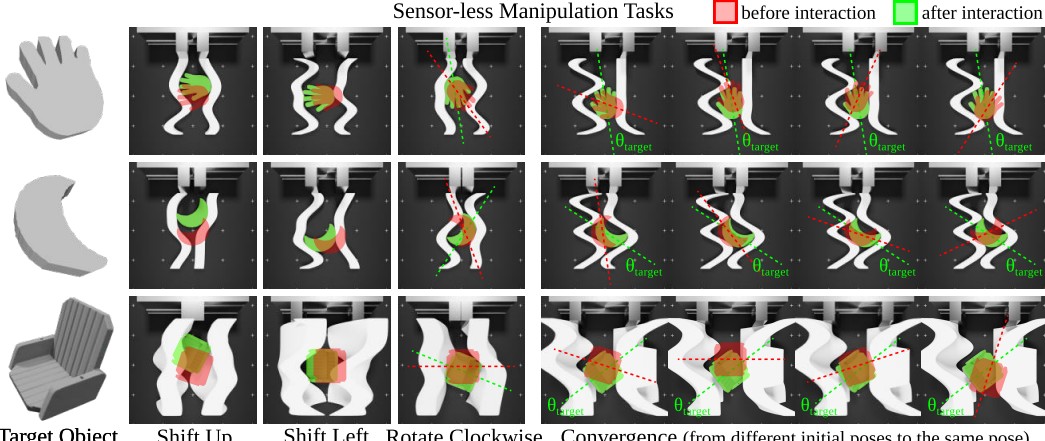

Figure 1: **Task-specific Designs without Task-specific Training.** Given different input objects (1st column), DGDM generates diverse manipulator geometries tailored to different manipulation tasks without task-specific training, which can be deployed under the sensor-less setting with an open-loop parallel closing motion.

## 1 Introduction

Mechanical intelligence refers to the utilization of mechanical design to solve tasks [1]. A substantial body of evidence in both natural [2] and artificial systems [3] has demonstrated that well-customized embodiments can significantly simplify an agent's perception and control, thereby enhancing overall robustness [4]. Despite its advantages, mechanical intelligence in robotics has recently been overshadowed by the rapid development of its counterpart, "action intelligence", where the agent focuses on inferring different actions for different tasks, assuming a fixed mechanical embodiment design.

In contrast, learning for mechanical design has largely focused on single task optimization [5, 6] or heavily engineered objective functions that could not be reused for new design task [7, 8, 9, 10, 11,

8th Conference on Robot Learning (CoRL 2024), Munich, Germany.

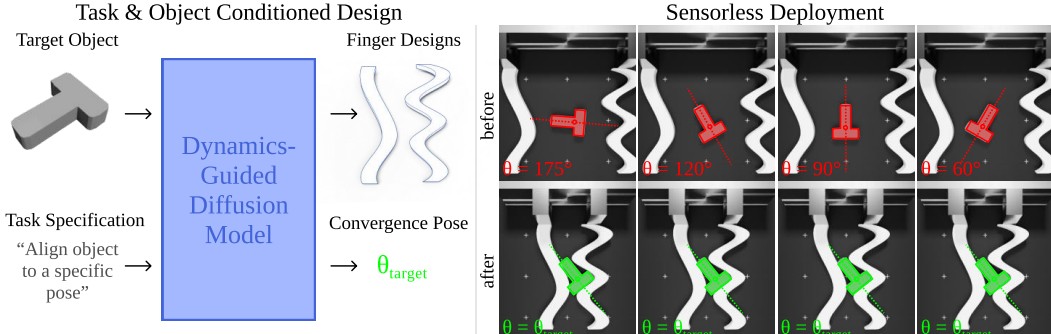

Figure 2: **The Convergence Task** is to design fingers that always reorient a target object to a specified orientation $\theta_{target}$ (in the manipulator frame) when closing the gripper in parallel. This task enables funneling objects from arbitrary poses to a specific $\theta_{target}$ in a sensor-less setting, and moving objects to any particular configuration combined with a global transformation of the gripper. Despite its utility, designing for convergence can be counter-intuitive – it often takes an expert many design cycles to come up with just one design for one object. In contrast, DGDM can generate a functional design for a new object in seconds.

12]. In practice, this means automating task-specific design typically involves recollecting training data for every scenario, which is too expensive to be practical. Therefore, we investigate the following question: *Can we automate task-specific mechanical design without task-specific training?*

We introduce **Dynamics-Guided Diffusion Model**, a framework that generates manipulator geometry designs that can manipulate objects towards desired motions and poses with no task-specific training and no perception or closed-loop control - only a parallel jaw closing motion. From tasks as simple as shifting/rotating objects to complex tasks requiring sequential interactions such as pose convergence (Fig. 2), DGDM generates designs in seconds with geometry changes that are highly adapted to the task and object. Our framework answers two key research questions:

- **How to represent the task space?** The task representation has to be expressive enough to capture the wide range of manipulation tasks while being compact enough to be readily learned from data. Our key insight is that many manipulation tasks can be decomposed into a collection of individual motion targets that specify how each object should move under each initial pose, which we call *interaction profile*. While the final composed objective is specific to the task, each of the individual motions can be modeled by a generic *dynamics network* that is reusable across tasks.
- **How to facilitate efficient search?** As the design space grows, the design objective landscape often becomes multi-modal w.r.t. the design parameters, and generating promising yet diverse design candidates becomes challenging. To address this issue, we first represent the design space using an unconditional *geometric diffusion model*. Then, the interaction profile for an object and the fingers is inferred with the dynamics network. The *design objective* constructed by comparing the current with the target interaction profiles gives us a gradient on how to update the finger. This *dynamics guidance* is incorporated into diffusion denoising steps similarly to classifier guidance [13].

We demonstrate results on both 2D and 3D objects with a variety of manipulation objectives ranging from simple to complex and single- to multi-object objectives, all under a sensor-less setting, where the initial pose of the object is unknown. Experiments in simulation and the real world demonstrate that designs generated by DGDM achieve high task performance, with 31.5% and 45.3% relative success rate improvements compared to optimization and unguided diffusion baselines.

## 2   Related work

**Manual End-effector Design.** The diverse array of manipulator designs we see today, including serial versus parallel, from dexterous to underactuated, typically start from many trail-and-error iterations by experts. Heavy manual efforts are needed to discover optimal and occasionally counter-intuitive designs, which hinders the development of designs for new applications. For instance, for complex manipulation tasks such as convergence (Fig. 2), previous works only deal with 2D planar polygons, utilizing manual/analytical designs of grasping policy [14, 15] or gripper geometry [16].

**Analytical Optimization for Automatic End-effector Design.** To alleviate the manual efforts, previous works have explored optimization approaches to manipulator design. Non-linear optimization approach [17] typically requires careful task-specific formulation of objectives and constraints. First-order optimization of morphology [18] or both morphology and control [17, 7, 19] is more common, but requires careful initialization (task-specific parameterization [18], cage-based deformation [7, 8], or heuristics [19]). Further, tasks involving complex contact modes are known to yield biased and high variance gradients in differentiable simulators [20, 8]. Importantly, all manual efforts involved in setting up an optimization problem are typically not transferrable to new tasks.

**Data-driven Robot Hardware Design.** Data-driven approaches improve over optimization-based approaches by transferring knowledge from training to reduce the cost at inference. A common approach is to train a value network that takes the design parameterization as input and outputs the design's task performance. This value network can be used to guide a search/optimization procedure, which has been explored in gripper design [5] and locomotion [21, 22], to guide optimization [5, 23, 6] or graph search [21, 22, 10, 24]. Another approach is to learn a generative model of the design space, which compresses the design space into a low-dimensional continuous latent space. This makes offline optimization via gradient-descent [5, 23] or online optimization via trial-and-error rollouts of random latent-space samples [5, 10, 6] significantly more efficient. Finally, when co-optimizing morphology and control, leveraging control experience from prior embodiment evaluations can significantly improve the efficiency and accuracy of new embodiment evaluations [9, 12]. However, all these approaches require a large amount of task-specific data, while we eliminate this requirement by leveraging dynamics as the shared structure between manipulation tasks.

## 3 Approach

### 3.1 Interaction Profiles as Task Specification

Requirements for a manipulation system are incredibly diverse, ranging in what initial poses are allowed, what objects are considered, and what the desired effects are. Parameterizing the space of manipulation tasks call for a representation expressive enough to capture all the diverse tasks. Moreover, this representation should be compact - containing only the necessary information to capture how the object interacts with the finger, such that it is efficient to evaluate/learn. For instance, modeling the detailed physics states in differentiable simulators [7] is expressive, but forward integrating the dynamics over the time horizon for every finger evaluation is expensive.

**Interaction Profiles.** Many manipulation tasks can be decomposed into a collection of individual motion targets that specify how each object should move under each initial pose after interacting with the manipulator. By combining motions from all objects and initial poses, we get a complete profile of how the manipulator will interact with the target objects - the "interaction profile".

Denote by $o$ and $m$ the geometry parameters of object shape and manipulator shape. When the object is at the initial planar pose $p = (\theta, x, y)$, closing the manipulator once will change the object's pose by $\Delta p = (\Delta \theta, \Delta x, \Delta y)$, dictated by the manipulator-object interaction dynamics $\mathcal{D}$. We refer to scalar-valued functions $f$ defined on top of $\Delta p$ as motion objectives and aggregate these motion objectives among all initial poses $p$ and objects $o$ to get the design objective $F$.

> **Example: Multi-object Shift Up**
>
> To design a manipulator that shifts a set of objects upwards, each motion objective is defined as
>
> $$f(o, m, p) = \Delta y(o, m, p) \tag{1}$$
>
> where $\Delta y$ is the y-translation component of $\Delta p$. The design objective is aggregated from (1) as
>
> $$F(m) = \sum_o \sum_p f(o, m, p) \tag{2}$$

Interaction profiles can scale to varying ranges of initial poses and objects. Thus, using a larger set of initial poses and objects will yield an objective that is more robust to different initial poses and

tailored to more objects. Since each motion objective is conditioned on $p$, this approach also allows for objectives dependent on initial poses, as illustrated in the following example.

> **Example: Pose Convergence**
>
> The goal of pose convergence is to rotate an object to a target orientation $\theta_{\text{target}}$ relative to the manipulator (Fig. 2). The manipulator should funnel a wide range of initial configurations into a single target orientation with no perception, no closed-loop control, only a parallel gripper closing motion on repeat. How the object should rotate depends on the initial orientation $\theta$ relative to the target orientation $\theta_{\text{target}}$:
>
> $$f(o,m,p) = \begin{cases} \Delta\theta(o,m,p) & \text{if } \theta \in [\theta_{\text{target}} - \pi, \theta_{\text{target}}] \\ -\Delta\theta(o,m,p) & \text{if } \theta \in [\theta_{\text{target}}, \theta_{\text{target}} + \pi] \end{cases} \tag{3}$$
>
> The objective for this task can then be aggregated from (3) analogously to (2).

If $\nabla_m F$ can be efficiently computed, we can use this gradient to optimize a pair of fingers $m$ that achieves the task. To achieve this, we propose to represent interaction dynamics $\mathcal{D}$ as a neural network and train it using data generated from interactions between random finger-object pairs.

## 3.2 Dynamics Network

The dynamics network $\mathcal{D} : (o,m,p) \mapsto \Delta p$ aims to learn a general model of how a random distribution of fingers interacts with a distribution of objects. Importantly, it provides gradients of the design objective with respect to the finger representation (Fig. 3).

**Shape Representation.** We choose cubic Bézier curves and surfaces as the manipulator shape representation [25]. Control points are grid sampled along the length (and height in 3D) of the finger while the remaining y-coordinate determines its protrusion outwards/inwards, which we define as $m$ - the geometry parameter of manipulator. We represent object shape $o$ as a point cloud by sampling 100 points from each 2D object contour and 512 3D points from 3D object surfaces.

**Motion Representation.** We represent object motion under interaction as a three-dimensional vector consisting of delta rotation along the z-axis, delta translation along the x-axis, and delta translation along the y-axis, denoted as $\Delta p = (\Delta\theta, \Delta x, \Delta y)$.

**Network Architecture.** First, we transform object initial poses $p$ with a high-frequency positional encoding - a trick used to combat over smoothing of neural networks [26]. Then, $o$ and $m$ are passed through separate 2-layer MLPs with 256 hidden dimensions, before being concatenated with the pose embedding. Finally, the resulting embedding is passed through an 8-layer MLP with 256 hidden dimensions to get the predicted object motion $\Delta p$. In 3D, we use a PointNet++ [27] to encode object geometry, whereas all other parts of the network are shared between 2D and 3D tasks.

**Training Data Generation.** Our training data generation happens once for all tasks. We sample object and manipulator pairs, load them into MuJoCo [28] simulation environment, and measure $\Delta p$ after a single parallel closing interaction. We generate 321 planar object shapes from the Icons-50 dataset [29] in 2D, and select 164 objects from Google's Scanned Objects Dataset [30, 31] in 3D. We randomly sample 1024 manipulator geometry parameters $m$ from a uniform distribution. For each object-fingers pair, we grid sample 360 initial orientations and 25 initial positions, getting $321 \times 1024 \times 360 \times 25$ training data points for 2D dynamics network and $164 \times 1024 \times 360 \times 25$ for 3D.

**Design Objective Gradient Evaluation.** To design manipulators that are generalizable to more initial poses $p$ and objects $o$, the design objective $F$ (2) can be evaluated for a wider range of poses and objects. We grid-sample initial poses of a set of objects and evaluate motion objectives in parallel. The design objective gradient $\nabla_m F$ is attained by aggregating the gradients along the pose/object batch dimension. For each new design, evaluating $\nabla_m F$ takes 0.16 seconds on average, making it efficient to run in the inner loop of iterative design procedures. Specifically, we grid-sample 360 orientations and $5 \times 5$ positions, getting a $360 \times 5 \times 5$ dimensional motion profile.

## 3.3 Dynamics-Guided Diffusion Model

Given design objective gradients from $\mathcal{D}$, the obvious approach is to perform gradient descent on the finger geometry [5, 19]. However, the distribution of good designs are often multi-modal, which means gradient descent approaches quickly get stuck in local minima. To efficiently navigate through the large and multi-modal design space, we extend classifier guidance [13], an iterative diffusion model sampling approach that enables a balance between diversity and task-specific guidance (Fig. 3).

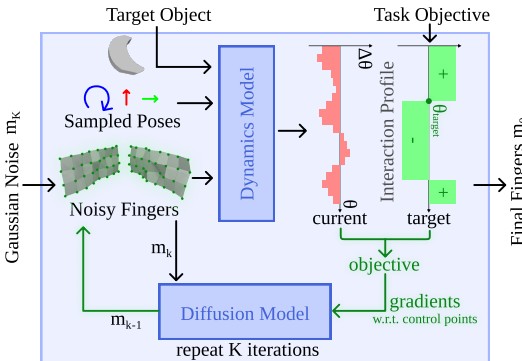

Figure 3: **DGDM** generates finger shapes given a target object and task, specified as a target interaction profile (§ 3.1). This is compared with the dynamics network's prediction of the current interaction profile, which is used to construct an objective (§ 3.2). Gradients of the objective iteratively guide the reverse denoising process of a manipulator shape diffusion model (§ 3.3).

**Diffusion Models** [32, 33] are a class of probabilistic generative models that generate samples from an underlying distribution through iterative denoising. A diffusion model $\varepsilon_\theta(m_k)$ predicts the noise added to a sample $m_0$. We start with a Gaussian noise $m_K$ and gradually predict less-noisy samples $m_{K-1}, m_{K-2}, \ldots$ until $m_0$ through a reverse noising process of modeling the distribution $p_\theta(m_{k-1}|m_k)$. Specifically, we sample geometry parameters of manipulator $m$ from a uniform distribution and train a *geometric diffusion model* (with 1D UNet architecture [34]) on this distribution once and for all tasks. We employ Denoising Diffusion Implicit Models (DDIMs) [35] for diffusion sampling process with 15 training denoising iterations and 5 inference iterations, and the Square Cosine noise scheduler [36].

**Classifier Guidance** [13] guides the reverse noising process with priors of an unconditional diffusion model. It requires a classifier $p_\phi(l|m_k)$, where $m_k$ is the sample, $l$ is the class label, and $\phi$ is the classification network. Leveraging the connection between diffusion models and score matching [37, 38], a new noise prediction can be defined as:

$$\hat{\varepsilon}(m_k) := \varepsilon_\theta(m_k) - \sqrt{1 - \bar{\alpha}_k} \nabla_{m_k} \log p_\phi(l|m_k) \tag{4}$$

where $\bar{\alpha}_k := \prod_{t=1}^{k} 1 - \beta_t$, $\beta_t$ is the variance of Gaussian noise added to samples at step $t$. Then DDIM can be performed with the modified noise prediction for conditioned sampling.

**Dynamics Guidance.** To guide the design generation towards specified manipulation tasks, we extend classifier guidance to use design objectives constructed from interaction profiles, which we term dynamics guidance. We replace classifier gradients with $\nabla_{m_k} F(m_k)$ to guide the DDIM sampling process (Algo. 1). We not only enable guiding unconditional diffusion models with task-specific gradients, but also allow tuning the guidance scale $s$ to trade off diversity and performance. Dhariwal and Nichol [13] showed that

---

**Algorithm 1** Dynamics-guided DDIM sampling, given a diffusion model $\varepsilon_\theta(m_k)$, design objective $F(m_k)$, and gradient scale $s$.

---

Input: design objective $F(\cdot)$, gradient scale $s$
$m_K \leftarrow$ sample from $\mathcal{N}(0, \mathbf{I})$
**for all** $k$ from $K$ to 1 **do**
   $\hat{\varepsilon} \leftarrow \varepsilon_\theta(m_k) - s\sqrt{1 - \bar{\alpha}_k} \nabla F(m_k)$
   $m_{k-1} \leftarrow \sqrt{\bar{\alpha}_{k-1}} \left( \frac{m_k - \sqrt{1 - \bar{\alpha}_k}\hat{\varepsilon}}{\sqrt{\bar{\alpha}_k}} \right) + \sqrt{1 - \bar{\alpha}_{k-1}} \hat{\varepsilon}$
**end for**
**return** $m_0$

---

when $s$ is larger the distribution becomes sharper and generated samples have higher fidelity, while smaller $s$ leads to more diverse samples, which we also observed in our generated results (Fig. 4).

## 4 Evaluation

**Manipulation Tasks & Metrics.** We evaluated each approach on held-out objects (8 in 2D, 6 in 3D) and manipulation tasks. Each pair of fingers is mounted to a WSG50 gripper performing a fixed open-close action. We categorize our suite into two difficulty levels: 1) **Simple objectives** involve single-axis object movements in *SE2* space, including shifting up/down/left/right and rotating clockwise/counterclockwise. For each object-manipulator pair, we grid-sampled 360 planar initial object orientations and performed fixed open-close actions. The task is considered as successful if the movement of the object along the specified axis is larger than a predefined threshold (0.03 rad for rotation, 3 mm for x-axis translation, 2 mm for y-axis translation). For example, a manipulator

designed for the rotating objective succeeds if it rotates the object larger than 0.03 rad after the first closing action. Then, we report the average success rate over all sampled initial object orientations. 2) **Complex objectives** combine multiple simple objectives to parameterize a broad range of manipulation tasks. For the convergence objective, we report the *maximum convergence range* (°), indicating the broadest range of initial object orientations that can be driven towards a consistent final orientation within a small tolerance (5°). Observing continued object movement, we report the metric after the 40th open-close manipulator action. Additionally, we explored rotate clockwise and shift up/left, and rotate either way objectives to showcase DGDM's flexibility in composing conflicting objectives. These tasks were evaluated on average success rates.

**Comparisons.** Removing the dynamics guidance yields the **Unguided** baseline, which generates task-agnostic manipulators using our geometric diffusion model. Removing our diffusion model yields the **GD** baseline, which optimizes the manipulator control points using gradients of the design objective $\nabla F$ via gradient descent optimization, common for many prior works [5, 7, 8, 19]. We also evaluated a gradient-free optimization baseline - **CMA-ES** [39] (covariance matrix adaptation evolution strategy) that optimizes finger control points from objectives constructed from the dynamics network. Notably, we ran it with more than ×10 the compute (and ×10 time) of DGDM. To mitigate performance variance due to initialization, we ran each approach 16 times with different initializations per object-task pair and selected the best performance, then averaged among objects.

## 4.1 Experiment Results

**Generating task-specific manipulators designs.** DGDM generates tailored fingers for a wide variety of scenarios, surpassing the unguided baseline across all tasks. The advantages of generating custom fingers become more pronounced as the design requirements escalate. For instance, DGDM exhibited a +16.6% improvement over the unguided baseline in 2D simple objectives, a figure that expanded to 20.0% in 2D complex objectives

Table 1: **Single Object Evaluation** (Avg success rates % and convergence range °).

| | | Simple Objective | | | | | | Complex Objective | | | |
|---|---|---|---|---|---|---|---|---|---|---|---|
| | | up | down | left | right | clock | counter | rotate | clockup | clockleft | converge |
| 2D | Unguided | 56.8 | 82.1 | 82.9 | 80.4 | 46.9 | 58.5 | 74.0 | 36.4 | 36.9 | 61.7° |
| | GD | 79.5 | 53.3 | 81.3 | 94.0 | 48.8 | 73.2 | 78.9 | 29.3 | 49.4 | 73.7° |
| | CMA-ES | 80.7 | 82.2 | 88.1 | 97.0 | 60.5 | **73.9** | **80.3** | 52.2 | 55.8 | 73.4° |
| | DGDM | **88.2** | **92.0** | **96.7** | **97.7** | **60.8** | 72.0 | 79.3 | **62.8** | **63.7** | **83°** |
| 3D | Unguided | 43.0 | 43.8 | 80.4 | 87.9 | 41.2 | 33.5 | 64.2 | 30.3 | 33.1 | 63.6° |
| | GD | 47.4 | 66.3 | 86.3 | 88.1 | 59.1 | 52.0 | 66.9 | 29.2 | 37.7 | 60° |
| | CMA-ES | 50.2 | 70.8 | 85.2 | 80.9 | 53.9 | 50.3 | 72.7 | 32.7 | 42.1 | 70° |
| | DGDM | **81.5** | **75.1** | **95.1** | **97.2** | **69.9** | **65.0** | **83.0** | **57.1** | **58.2** | **72.5°** |

(see Tab. 1). A similar trend was observed when transitioning from 2D to 3D objects (+18.0% over Unguided in 2D, +23.4% over Unguided in 3D, Tab. 1) and from single-object to multi-object designs (+18.0% over Unguided in single-obj 2D, +18.6% over Unguided in multi-obj 2D, Tab. 2).

When task complexity, design space, and the target object set grow, a human expert designer would face significantly increased time and effort. DGDM handles progressively complex design requirements by aggregating gradients from individual motion objectives. A new task can be specified as long as users can articulate how *each* object should move from *each* initial pose, and can be seamlessly incorporated into the diffusion denoising process.

Table 2: **Multi-object Evaluation**

| | | Simple | | | | | | Complex | | |
|---|---|---|---|---|---|---|---|---|---|---|
| | | up | down | left | right | clock | counter | rotate | clockup | clockleft |
| 2D | Unguided | 55.8 | 79.8 | 77.1 | 80.2 | 44.7 | 56.4 | 68.3 | 35.2 | 35.2 |
| | GD | 78.6 | 50.3 | 79.2 | 93.8 | 46.1 | **71.4** | 74.3 | 25.0 | 49.0 |
| | CMA-ES | 77.7 | 62.2 | 79.3 | 97 | 51.8 | 70.6 | 73.1 | 22.1 | 37.3 |
| | DGDM | **83.8** | **88.1** | **99.3** | **94.3** | **61.3** | 68.4 | **78.4** | **62.4** | **63.8** |
| 3D | Unguided | 40.1 | 40.8 | 75.8 | 87.9 | 34.7 | 29.2 | 61.7 | 29.2 | 25.2 |
| | GD | 42.4 | 66.4 | 77.9 | 86.6 | 40.3 | 39.2 | 67.0 | 22.6 | 34.3 |
| | CMA-ES | 50.4 | 65.6 | 76.9 | 89.2 | 45.2 | 41.9 | 68.4 | 28.7 | 33.5 |
| | DGDM | **89.7** | **66.8** | **96.1** | **95.4** | **69.3** | **58.2** | **77.6** | **44.2** | **37.9** |

**Robust & efficient search with guided diffusion.** The GD baseline and DGDM share the same design objective gradients from our dynamics network, differing only in how this gradient information is incorporated. The baseline uses gradient descent, requiring upwards of 18 and 24 minutes to converge in 2D and 3D (for 16 samples), respectively, and is prone to local minima. In contrast, DGDM utilizes classifier-guidance with a diffusion denoising process, which strikes a balance between exploring different modes (by introducing Gaussian noise) and exploiting the current mode (using the gradient of design objective) through a guidance scaling factor (Fig. 4). This results in +12.8% and +10.4% higher success rates

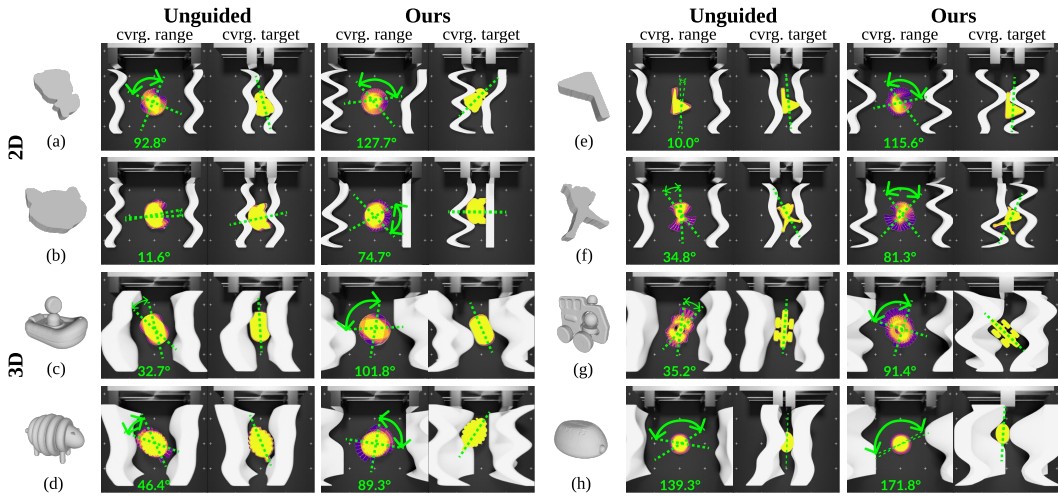

Figure 5: **Convergence Results.** For each pair of finger designs, we show the range of initial orientations ("cvrg. range") which converges to the same convergence mode ("cvrg. target").

than the baseline in 2D and 3D simple objectives, respectively. Additionally, our diffusion models prove stable even with diffusion processes as short as 5 timesteps, translating to an average design time of 13 and 54 seconds in 2D and 3D, respectively.

**Emergent design for convergence.** What strategies do our designs employ to achieve convergence from a broader range of initial orientations compared to the unguided baseline (Tab. 1)? We identify two emergent design patterns: 1) **Push-and-Catch**: One finger features a bulge that pushes the object into the hollow cavity of the other finger (Fig. 5 a,b,d-f,h). This cavity roughly complements the object's shape at the convergence point. 2) **Parallel-Align**: When objects exhibit symmetric flat edges, our

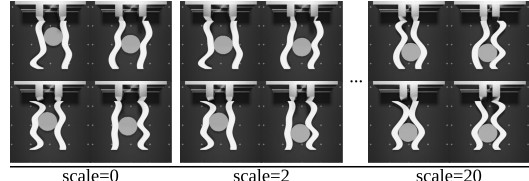

Figure 4: **Effect of Scaled Guidance.** From left to right we increase the scaled guidance in the diffusion process. The increased scale enforces more task guidance and achieves higher task performance (shifting down) while reducing the diversity of generated designs.

designs utilize two parallel surfaces to align these edges (Fig. 5 c,g). The generated designs simultaneously exploit object geometry and physics to achieve the most effective convergence.

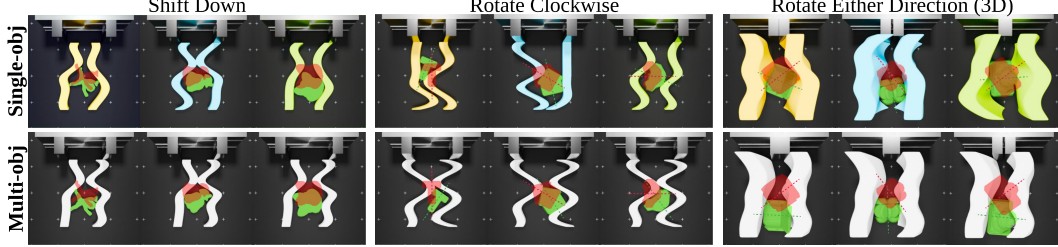

Figure 6: **Specialized or Generalized Design in Multi-object Scenarios.** Our approach can be flexibly conditioned on individual objects and generate a specialized design for each object [Top] or simultaneously conditioned on multiple objects and generate one design for all objects [Bottom].

**Specialized or generalized designs for multi-object scenarios.** DGDM is able to generate more generic designs for multiple objects (Fig 6). In contrast, the unguided baseline lacks task-specific guidance, hindering its ability to guide its generations toward a common design objective for all objects. On the other extreme, the GD baseline often gets stuck in a local minimum. These limitations are reflected in Tab. 2, with our approach achieving +18.6% and +23.4% higher success rates than the unguided baseline, and +14.7% and +17.6% higher success rates than the GD baseline. Naturally, we acknowledge that multi-object finger designs often sacrifice some performance compared to the single-object scenario (−1.5% in 2D, −5.2% in 3D). This balance between generality and

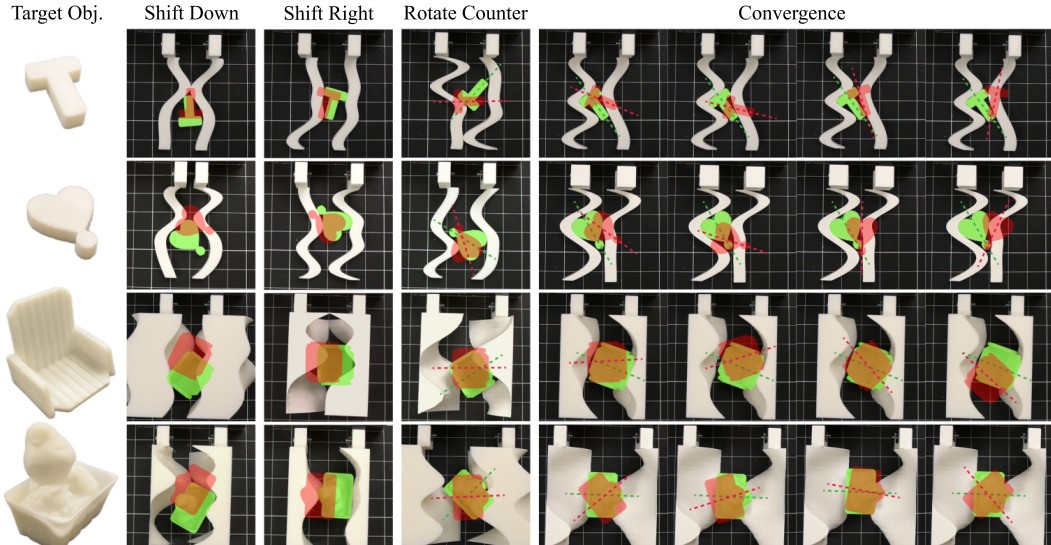

| Target Obj. | Shift Down | Shift Right | Rotate Counter | Convergence |

Figure 7: **Real-world Results.** We manufacture manipulators generated by DGDM and execute the open-loop parallel closing motion. Behaviors in simulation successfully transfer to the real world. Red and green masks denote object configurations before and after interaction respectively.

task-specific performance is a fundamental trade-off in mechanism design in automation, with the optimal compromise dependent on the specific application.

**Real-world evaluation with Sim2Real transfer.** We show real-world results of all tasks for both 2D and 3D cases by mounting the 3D printed designs on a WSG50 gripper (Fig. 7). The material we use for 3D printing objects and fingers is PLA and the molding solution is FDM. In Tab. 3 we show the real-world and simulation quantitative results side by side. For the shifting down/right and rotating counterclockwise tasks, we tested with 10 random initial poses (0°-360° orientations and ±5 cm from the center) of the object in the real world and reported the average success rate (%). For the convergence task, the maximum range of initial orientations (°) that can be driven to the target convergence pose is reported. We observe that the performance in the real world is very close and oftentimes better than the simulation result, suggesting a small sim2real gap.

This is due to our sensor-less formulation, where we do not need perception or closed-loop control. With the same fixed parallel motion in sim and real, the transferability is only determined by the geometry and contact physics. Our dynamics guidance is conditioned on many initial object poses, enabling the generated designs to be robust to different initial object poses, relying on more prominent physical phenomena

Table 3: **Real-world Quantitative Results**

|    |        | Shift Down | | Shift Right | | Rotate Counter | | Convergence | |
|    |        | sim | real | sim | real | sim | real | sim | real |
|----|--------|-----|------|-----|------|-----|------|------|------|
| 2D | T      | 93.1 | 90  | 100 | 100  | 75.3 | 70  | 111° | 124° |
|    | Heart  | 78.3 | 80  | 99.2 | 100 | 66.1 | 70  | 117° | 119° |
| 3D | Chair  | 91.4 | 100 | 100 | 100  | 68.6 | 70  | 93°  | 86°  |
|    | Basket | 96.9 | 100 | 100 | 100  | 65.3 | 70  | 82°  | 88°  |

that are consistent between sim and real. Moreover, the sensor-less manipulation tasks only require the directions of individual object motions to be accurate but not the magnitudes. We see this effect when we use PLA with a lower friction coefficient in real than in sim, allowing the objects to slide more smoothly but in the same direction, leading to oftentimes better performance in real.

## 5   Conclusions

We present Dynamics-Guided Diffusion Model, a versatile framework for the rapid generation of diverse and tailored manipulator geometry designs for unseen tasks. With the ability to generate a new design within 0.8s, this task-agnostic framework lays the groundwork to enable more rapid experimentation and future research. We hope that our framework contributes to the wider adoption of data-driven approaches in robotic mechanism design.

## Acknowledgements

We thank Zhenjia Xu, Cheng Chi, Mandi Zhao, Zeyi Liu, Yifan Hou, Austin Patel, Chuer Pan, Yihuai Gao, Dominik Bauer, Samir Gadre, Mengda Xu and John So for their thoughtful discussions and helpful feedback on initial drafts of the manuscript. This work was supported in part by the NSF Award #2143601, #2037101, and #2132519. We would like to thank Google for the UR5 robot hardware. The views and conclusions contained herein are those of the authors and should not be interpreted as necessarily representing the official policies, either expressed or implied, of the sponsors.

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

# Appendix

## 1 Manipulation tasks and metrics

We provide more details about the evaluation manipulation tasks, and introduce more metrics for each task in this section.

Simple objectives:

- **Shift up**: Shift the object upward (along the negative x-axis in the manipulator frame) under all initial poses. For evaluation, we grid sample 360 initial object orientations with positions in the center of the manipulator, and close the manipulator around the object. The average success rate after one closure $S_x^- \uparrow (\%)$ is reported, where success is counted if the object is shifted more than 3 mm along the negative x-axis. We additionally report continuous metrics including average delta object translation along the x-axis after one closure $\Delta x \downarrow$ (cm), and average final x coordinate of the object after the 40th gripper closure $x \downarrow$ (cm). The metrics are averaged among the 360 interaction trails with different initial object orientations. The motion objective is $f(o,m,p) = -\Delta x(o,m,p)$, then the design objective is aggregated from $f(o,m,p)$ as $F(m) = \sum_o \sum_p f(o,m,p)$.
- **Shift down**: Shift the object downward (along the positive x-axis) under all initial poses. Metrics are average success rate of the object being shifted more than 3 mm along the positive x-axis after one closure $S_x^+ \uparrow (\%)$, average delta translation along the x-axis $\Delta x \uparrow$ (cm), and average final x coordinate $x \uparrow$ (cm). The motion objective is $f(o,m,p) = \Delta x(o,m,p)$.
- **Shift left**: Shift the object leftward (along the negative y-axis) under all initial poses. Metrics are average success rate of the object being shifted more than 2 mm along the negative y-axis after one closure $S_y^- \uparrow (\%)$, average delta translation along the y-axis $\Delta y \downarrow$ (cm), and average final y coordinate $y \downarrow$ (cm). The motion objective is $f(o,m,p) = -\Delta y(o,m,p)$.
- **Shift right**: Shift the object rightward (along the positive y-axis) under all initial poses. Metrics are average success rate of the object being shifted more than 2 mm along the positive y-axis after one closure $S_y^+ \uparrow (\%)$, average delta translation along the y-axis $\Delta y \uparrow$ (cm), and average final y coordinate $y \uparrow$ (cm). The motion objective is $f(o,m,p) = \Delta y(o,m,p)$.
- **Rotate clockwise**: Rotate the object clockwise (negative delta rotation around the z-axis) under all initial poses. Metrics are average success rate of the object being rotated more than 0.03 rad around the negative z-axis after one closure $S_\theta^- \uparrow (\%)$, average delta rotation around the z-axis $\Delta \theta \downarrow (°)$, and average final orientation $\theta \downarrow (°)$. The motion objective is $f(o,m,p) = -\Delta\theta(o,m,p)$.
- **Rotate counterclockwise**: Rotate the object counterclockwise (positive delta rotation around the z-axis) under all initial poses. Metrics are average success rate of the object being rotated more than 0.03 rad around the positive z-axis after one closure $S_\theta^+ \uparrow (\%)$, average delta rotation around the z-axis $\Delta \theta \uparrow (°)$, and average final orientation $\theta \uparrow (°)$. The motion objective is $f(o,m,p) = \Delta\theta(o,m,p)$.

Complex objectives:

- **Rotate**: Rotate the object either clockwise or counterclockwise under all initial poses. The metrics are average success rate $S_\theta^{+-} \uparrow (\%)$, the average absolute value of delta rotation around the z-axis $|\Delta\theta| \downarrow (°)$, and the average absolute value of final orientation $|\theta| \downarrow (°)$. The motion objective is $f(o,m,p) = [\Delta\theta(o,m,p)]^2$.
- **Rotate clockwise and shift up**: Rotate the object clockwise and shift it up under all initial poses. The metrics are average success rate $S_\theta^- \& S_x^- \uparrow (\%)$, average delta rotation around the z-axis $\Delta\theta \downarrow (°)$, average final orientation $\theta \downarrow (°)$, average delta translation along the x-axis after one closure $\Delta x \downarrow$ (cm), and average final x coordinate $x \downarrow$ (cm). The motion objective is $f(o,m,p) = -\Delta\theta(o,m,p) - \Delta x(o,m,p)$.
- **Rotate clockwise and shift left**: Rotate the object clockwise and shift it left under all initial poses. The metrics are average success rate $S_\theta^- \& S_y^- \uparrow (\%)$, average delta rotation around the z-axis $\Delta\theta \downarrow (°)$, average final orientation $\theta \downarrow (°)$, average delta translation along the y-axis $\Delta y \downarrow$ (cm), and average final y coordinate $y \downarrow$ (cm). The motion objective is $f(o,m,p) = -\Delta\theta(o,m,p) - \Delta y(o,m,p)$.

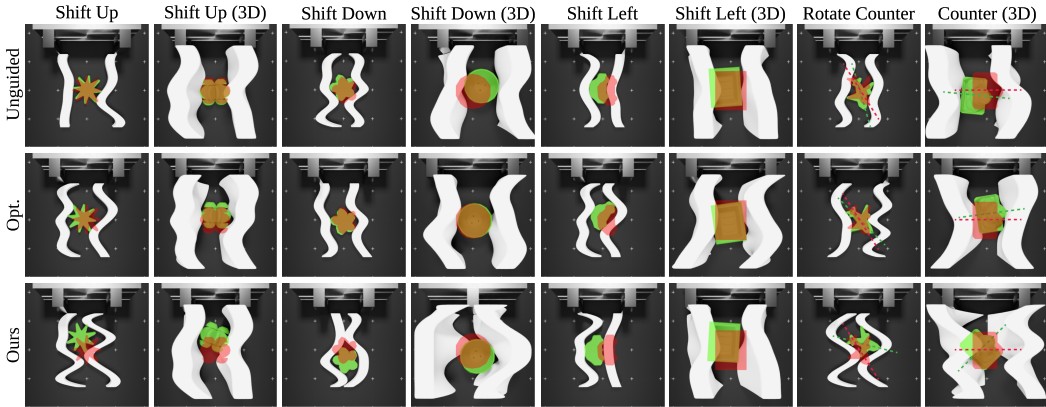

Figure 8: **Results on Simple Objectives.** We generate manipulators for simple objectives that involve motion along one dimension in $SE2$ space. Red and green object masks denote object configurations before and after interaction respectively, overlaid with the image after closing in the manipulator. Compared with baseline methods, finger shapes produced by our method achieve the task much more effectively.

- **Convergence**: Reorient the object towards a fixed final pose under a range of initial poses. The design objective is encouraging the object to rotate in the positive direction when its initial orientation is smaller than the target orientation, or else rotate in the negative direction. The motion objective is:

$$f(o,m,p) = \begin{cases} \Delta\theta(o,m,p) & \text{if } \theta \in [\theta_{\text{target}} - \pi, \theta_{\text{target}}] \\ -\Delta\theta(o,m,p) & \text{if } \theta \in [\theta_{\text{target}}, \theta_{\text{target}} + \pi] \end{cases} \tag{5}$$

To determine the best target orientation $\theta_{\text{target}}$, we first forward the dynamics network with the manipulator shape initialization, object shape, and sampled initial object poses to get a pseudo interaction profile. We detect the initial object orientation ranges that lead to consecutive positive delta rotations followed by consecutive negative delta rotations as pseudo convergence ranges. Then we select the largest pseudo convergence range's corresponding convergence orientation as $\theta_{\text{target}}$. The metric is the maximum convergence range $R_c^{max} \uparrow (°)$, the largest range of initial orientations leading to $\theta_{\text{target}}$ within a small tolerance. We report the maximum convergence range within the tolerance of 3°, 5°, and 10°, respectively.

## 2 Additional results

### 2.1 More metrics

We report results on all the manipulation tasks and metrics described above in Tab. 4, Tab. 5, Tab. 6, and Tab. 7. The evaluation procedure is the same as described in the main paper, where we run each approach 16 times per object-task pair and select the best performance, then average among test objects. DGDM outperforms baselines consistently on both discrete metrics (*e.g.* average success rate) and continuous metrics (*e.g.* delta object transformation and final transformation).

Table 4: Single Object Simple Objectives Evaluation

| | | up | | | down | | | left | | | right | | | clock | | | counter | | |
|---|---|---|---|---|---|---|---|---|---|---|---|---|---|---|---|---|---|---|---|---|
| | | $S_x^- \uparrow$ | $\Delta x \downarrow$ | $x \downarrow$ | $S_x^+ \uparrow$ | $\Delta x \uparrow$ | $x \uparrow$ | $S_y^- \uparrow$ | $\Delta y \downarrow$ | $y \downarrow$ | $S_y^+ \uparrow$ | $\Delta y \uparrow$ | $y \uparrow$ | $S_\theta^- \uparrow$ | $\Delta\theta \downarrow$ | $\theta \downarrow$ | $S_\theta^+ \uparrow$ | $\Delta\theta \uparrow$ | $\theta \uparrow$ |
| 2D | Unguided | 56.8 | -0.2 | -1.3 | 82.1 | 0.3 | 1.9 | 82.9 | -0.5 | -1.2 | 80.4 | 0.5 | 1.4 | 46.9 | -1.5 | -9.5 | 58.5 | 2.3 | 11.1 |
| | Opt. | 79.5 | -0.4 | -2.2 | 53.3 | 0.4 | 2.3 | 81.3 | -0.6 | -2.4 | 94.0 | 0.7 | 1.7 | 48.8 | **-2.9** | -8.8 | 73.2 | 3.6 | 9.6 |
| | DGDM | **88.2** | **-0.4** | **-3.1** | **92.0** | **0.5** | **3.7** | **96.7** | **-0.7** | **-2.4** | **97.7** | **0.8** | **2.1** | **60.8** | -2.6 | **-12.3** | **72.0** | **3.6** | **14.2** |
| 3D | Unguided | 43.0 | -0.1 | -0.6 | 43.8 | 0.1 | 1.0 | 80.4 | -0.2 | -1.2 | 87.9 | 0.2 | 0.9 | 41.2 | -1.1 | -4.9 | 33.5 | 0.5 | 2.7 |
| | Opt. | 47.4 | -0.1 | -1.3 | 66.3 | 0.2 | 1.7 | 86.3 | -0.3 | -1.3 | 88.1 | 0.3 | 1.6 | 59.1 | -1.9 | -5.0 | 52.0 | 1.2 | 4.6 |
| | DGDM | **81.5** | **-0.2** | **-1.6** | **75.1** | **0.2** | **1.7** | **95.1** | **-0.4** | **-1.8** | **97.2** | **0.4** | **1.9** | **69.9** | **-2.3** | **-7.7** | **65.0** | **2.2** | **6.3** |

Table 5: Single Object Complex Objectives Evaluation

| | | rotate | | | clock-up | | | | | clock-left | | | | | convergence | | |
|---|---|---|---|---|---|---|---|---|---|---|---|---|---|---|---|---|---|
| | | $S_\theta^{+-}$ ↑ | $|\Delta\theta|$ ↑ | $|\theta|$ ↑ | $S_\theta^-\&S_x^-$ ↑ | $\Delta\theta$ ↓ | $\theta$ ↓ | $\Delta x$ ↓ | $x$ ↓ | $S_\theta^-\&S_y^+$ ↑ | $\Delta\theta$ ↓ | $\theta$ ↓ | $\Delta y$ ↓ | $y$ ↓ | $R_c^{max}(3°)$ ↑ | $R_c^{max}(5°)$ ↑ | $R_c^{max}(10°)$ ↑ |
| 2D | Unguided | 74.0 | 3.5 | 18.2 | 36.4 | -1.5 | -9.5 | -0.2 | -1.3 | 36.9 | -1.5 | -9.5 | -0.5 | -1.2 | 56.5 | 61.7 | 68.7 |
| | Opt. | 78.9 | 4.5 | 18.9 | 29.3 | -1.8 | -4.2 | -0.3 | -1.0 | 49.4 | -2.9 | -9.0 | -0.6 | -2.1 | 69.6 | 73.7 | 82.1 |
| | DGDM | **79.3** | **4.5** | **20.1** | **62.7** | **-3.3** | **-14.5** | **-0.4** | **-3.6** | **63.7** | **-3.2** | **-10.1** | **-0.7** | **-2.4** | **78.4** | **83** | **89.3** |
| 3D | Unguided | 64.2 | 2.2 | 16.5 | 30.3 | -1.1 | -4.9 | -0.1 | -0.6 | 33.1 | -0.5 | -2.7 | -0.2 | -1.2 | 52.2 | 63.6 | 67.8 |
| | Opt. | 66.9 | 2.4 | 15.7 | 29.2 | -1.1 | -2.7 | -0.1 | -0.5 | 37.7 | -1.3 | -3.3 | -0.3 | -1.0 | 50.3 | 60 | 72.3 |
| | DGDM | **83.0** | **3.1** | **21.0** | **57.1** | **-2.4** | **-6.8** | **-0.2** | **-0.9** | **58.2** | **-2.9** | **-11.1** | **-0.5** | **-1.7** | **68.8** | **72.5** | **81.5** |

Table 6: Multi-object Simple Objectives Evaluation

| | | up | | | down | | | left | | | right | | | clock | | | counter | | |
|---|---|---|---|---|---|---|---|---|---|---|---|---|---|---|---|---|---|---|---|
| | | $S_x^-$ ↑ | $\Delta x$ ↓ | $x$ ↓ | $S_x^+$ ↑ | $\Delta x$ ↑ | $x$ ↑ | $S_y^-$ ↑ | $\Delta y$ ↓ | $y$ ↓ | $S_y^+$ ↑ | $\Delta y$ ↑ | $y$ ↑ | $S_\theta^-$ ↑ | $\Delta\theta$ ↓ | $\theta$ ↓ | $S_\theta^+$ ↑ | $\Delta\theta$ ↑ | $\theta$ ↑ |
| 2D | Unguided | 55.8 | -0.2 | -1.3 | 79.8 | 0.3 | 1.3 | 77.1 | -0.5 | -1.1 | 80.3 | 0.5 | 1.3 | 44.7 | -1.5 | -3.5 | 56.4 | 2.1 | 6.2 |
| | Opt. | 78.6 | -0.4 | -1.0 | 50.3 | 0.3 | 1.0 | 79.2 | -0.6 | -1.0 | 93.8 | 0.7 | 1.7 | 46.1 | **-2.9** | -7.8 | 71.4 | **3.5** | 7.4 |
| | DGDM | **83.8** | **-0.4** | **-3.0** | **88.1** | **0.4** | **3.4** | **99.3** | **-0.7** | **-2.5** | **94.3** | **0.7** | **2.1** | **61.3** | -2.4 | **-10.5** | **68.4** | 3.3 | **16.5** |
| 3D | Unguided | 40.1 | -0.1 | -0.5 | 40.8 | 0.1 | 0.9 | 75.8 | -0.2 | -0.8 | 87.9 | 0.2 | 0.6 | 34.7 | -0.5 | -0.8 | 29.2 | 0.1 | 2.5 |
| | Opt. | 42.4 | -0.1 | -0.4 | 66.4 | 0.2 | 1.0 | 77.9 | -0.2 | -0.4 | 86.6 | 0.3 | 0.8 | 40.3 | -1.1 | -1.9 | 39.2 | **1.9** | 3.5 |
| | DGDM | **89.7** | **-0.2** | **-1.5** | **66.8** | **0.2** | **1.2** | **96.1** | **-0.5** | **-1.8** | **95.4** | **0.4** | **1.3** | **69.3** | **-2.0** | **-5.2** | **58.2** | 1.6 | **3.5** |

Table 7: Multi-object Complex Objectives Evaluation

| | | rotate | | | clock-up | | | | | clock-left | | | | |
|---|---|---|---|---|---|---|---|---|---|---|---|---|---|---|
| | | $S_\theta^{+-}$ ↑ | $|\Delta\theta|$ ↑ | $|\theta|$ ↑ | $S_\theta^-\&S_x^-$ ↑ | $\Delta\theta$ ↓ | $\theta$ ↓ | $\Delta x$ ↓ | $x$ ↓ | $S_\theta^-\&S_y^+$ ↑ | $\Delta\theta$ ↓ | $\theta$ ↓ | $\Delta y$ ↓ | $y$ ↓ |
| 2D | Unguided | 68.3 | 3.2 | 13.4 | 35.2 | -1.5 | -3.5 | -0.2 | -0.8 | 35.2 | -1.0 | -3.0 | -0.5 | -1.0 |
| | Opt. | 74.3 | 3.8 | 7.2 | 25.0 | -1.4 | -0.1 | -0.1 | -0.2 | 49.0 | -2.9 | -4.6 | -0.4 | -0.7 |
| | DGDM | **78.4** | **4.2** | **15.8** | **62.4** | **-3.3** | **-10.6** | **-0.4** | **-3.6** | **63.8** | **-3.0** | **-6.3** | **-0.7** | **-2.4** |
| 3D | Unguided | 61.7 | 2.1 | 15.8 | 29.2 | -0.8 | -0.4 | -0.1 | -0.5 | 25.2 | 0.1 | 2.5 | -0.2 | -0.7 |
| | Opt. | 67.0 | 2.3 | 9.5 | 22.6 | 0.1 | -0.1 | -0.1 | -0.3 | 34.3 | -0.9 | -1.2 | -0.2 | -0.5 |
| | DGDM | **77.6** | **2.5** | **21.6** | **44.2** | **-1.6** | **-7.6** | **-0.2** | **-1.2** | **37.9** | **-2.3** | **-8.9** | **-0.5** | **-2.3** |

## 2.2 Results averaged over initializations

For each task and object shape, we generate manipulators with 16 different initializations. Aside from reporting the best of 16 trails in the main paper, we additionally report the average performance over all trails in Tab. 8. DGDM still significantly outperforms the baselines on metrics averaged over both different initialization and objects.

Table 8: Single object evaluation, metrics are **averaged** over different initializations and objects.

| | Simple Objective | | | | | | Complex Objective | | | |
|---|---|---|---|---|---|---|---|---|---|---|
| | up | down | left | right | clock | counter | rotate | clockup | clockleft | converge |
| Unguided | 13.3 | 24.5 | 27.3 | 26.4 | 22.4 | 26.8 | 49.2 | 5.4 | 9.7 | 38.4° |
| Opt. (GD) | 25.7 | 30.7 | 35.4 | 37.9 | 27.9 | 35.5 | 54.9 | 8.2 | 11.7 | 41.6° |
| CMA-ES | 34.0 | 42.6 | 41.5 | 51.8 | 32.8 | 38.0 | 56.2 | 16.6 | 16.2 | 40.5° |
| DGDM | **66.4** | **72.4** | **68.9** | **74.2** | **38.0** | **39.4** | **56.3** | **27.5** | **42.1** | **45.0°** |

## 2.3 Task progress over action horizon

We plot the progress of convergence over 40 open-close actions with manipulators generated by baselines and DGDM in Fig. 9. The Unguided baseline converges the slowest, taking more than 30 steps. The Opt. baseline exhibits an unstable dip at the start and only achieves consistent alignment with the target orientation after 30 steps. Manipulators generated by DGDM not only have larger convergence ranges but also converge faster (with 10 steps) than the baselines.

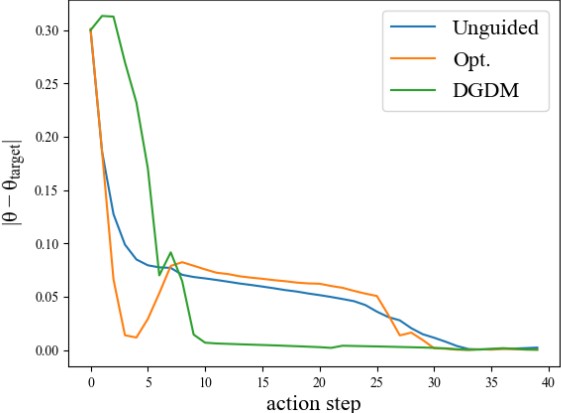

Figure 9: Task progress of convergence over 40 open-close actions. The vertical axis is $|\theta - \theta_{target}|$, denoting the absolute difference between the current object orientation with the target orientation.

## 3 More details on interaction profile

**Interaction profile.** The interaction profile between a manipulator and an object is defined as the distribution of the delta pose change of the object caused by the manipulator-object interaction over all possible initial poses of the object. Here, the manipulators-object interaction is a simple parallel closing action, and the object pose is represented as 2D translation and rotation. For example, in Fig. 10, when the object's initial pose is $(\theta, x, y)$, its delta pose after the interaction is $(\Delta\theta, \Delta x, \Delta y)$, which provide one data point on the interaction profile. Then, by uniformly sampling all initial poses, we obtain the complete interact profile.

In our implementation, we use simulation to obtain the **ground truth interaction profile** and train a dynamic network to infer the **predicted interaction profile** given any manipulator-object pair without simulation.

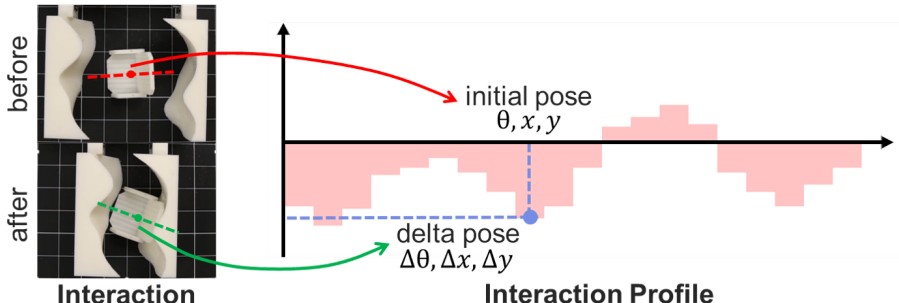

Figure 10: Interaction profile.

**Target interaction profile.** A target interaction profile is defined for a specific task objective. For example, the target interaction profile for the convergence task is shown in Fig. 11. This interaction profile indicates the object should rotate in a positive direction when its initial orientation is smaller than the convergence rotation; otherwise, it rotates in a negative direction. Tab. 9 shows examples of other target interaction profiles for different design objectives.

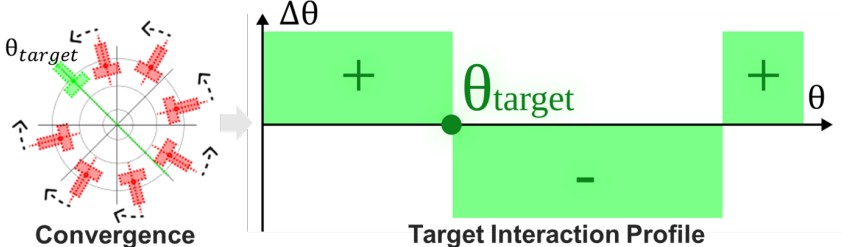

Figure 11: Target interaction profile.

**Target v.s. predicted interaction profile.** The difference between the target interaction profile and the predicted interaction profile (inferred by the dynamics network) provides the gradients for the manipulator design (Fig. 12). Since the dynamics network is fully differentiable, we can get its gradient with respect to the input manipulator geometry.

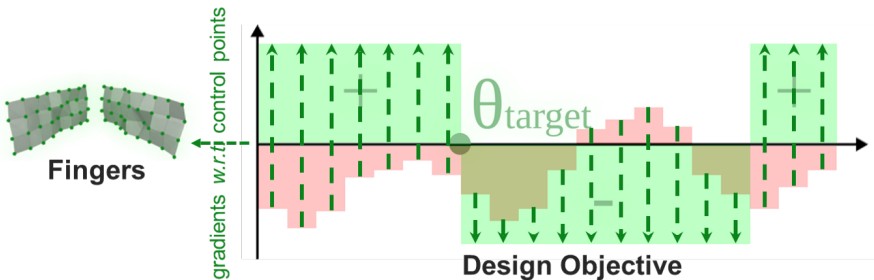

Figure 12: Target v.s. predicted interaction profile.

**List of target interaction profile plots.** We provide a comprehensive list of task objectives and their corresponding interaction plots in Tab. 9. For visualization purposes, we simplify the horizontal axes of interaction profile plots to include only initial orientations $\theta$.

Table 9: Target interaction profile plots

| Task | Objective Function $f(o,m,p)$ | Interaction Profile |
|------|-------------------------------|---------------------|
| converge | $\begin{cases} \Delta\theta(o,m,p) & \text{if } \theta \in [\theta_{\text{target}} - \pi, \theta_{\text{target}}] \\ -\Delta\theta(o,m,p) & \text{if } \theta \in [\theta_{\text{target}}, \theta_{\text{target}} + \pi] \end{cases}$ |  |
| up | $-\Delta x(o,m,p)$ |  |
| down | $\Delta x(o,m,p)$ |  |
| left | $-\Delta y(o,m,p)$ |  |
| right | $\Delta y(o,m,p)$ |  |
| clock | $-\Delta\theta(o,m,p)$ |  |
| counter | $\Delta\theta(o,m,p)$ |  |

## 4 Discussions

**Modeling Interactions instead of Modeling Contacts.** Differentiable simulator [7, 8, 6, 40, 41] is another popular choice for providing the gradient of design objective $\nabla_m F$, but suffers from two major limitations. First, soft contact models, such as penalty-based methods used by Xu et al., are known to yield biased and high-variance gradients [20, 8]. Further, such gradients need to be computed for each simulation timestep, which is computationally expensive for long-horizon interactions. Instead of modeling individual contacts, our dynamics network learns to capture the temporally extended finger-object interaction. It is trained on physically accurate simulated data, avoiding the limitations associated with soft contacts. Moreover, our dynamics network generalizes to novel objects and tasks at test time, allowing for constructing new objectives without extra data generation or training.

