# OpenReview forum: "Dynamics-Guided Diffusion Model for Sensor-less Robot Manipulator Design"
_robot-learning.org/CoRL/2024/Conference — CoRL 2024_

### Official Review · Reviewer_NVZK · 2024-07-20
**Good advancement in manipulator design with generative models, lacks some clarity and justifications**

**Originality:** 3
**Technical Quality:** 3
**Clarity Of Presentation:** 3
**Potential Impact:** 3
**Recommendation:** 2
**Confidence:** 4

**Review:**

The paper proposes Dynamics guided diffusion model to find optimal gripper finger geometry to perform certain tasks (like rotating and moving) on certain objects without any task-specific training or perception feedback. DGDM is based on a well established approach of that uses a classifier (here, a function of a learned dynamics model) in reverse diffusion to guide the sampling to satisfy an objective. Since the classifier can be flexibly used to specify task information at inference, training can be done in a more general way. Also, as the initial pose of the object is unknown (i.e. no sensors are used), the method does not require any sensory feedback and can generate a robust finger geometry design well suited for the task at hand. The method is evaluated for a variety of objects and tasks to show that DGDM works better than optimization based approaches and unguided diffusion version of itself. Also, DGDM is much faster than optimization based counterparts while achieving considerably higher success rates.

Strengths:

1. The work introduces a new generative model direction to manipulator design and will be a good motivation to explore this interesting direction more in the future.

Weakness:

1. While I understand the importance of the proposed method and application, I consider the work to be very limiting. While the authors agree that the motions are limited to 2D translation and rotation, but the utility of this approach in the shown assembly line situation is also based on lot of assumptions. For example, there is an upper limit on the permissible difference between current and target orientation for the manipulator to solve the task. Also, not all objects must be manipulated similarly, also agreeing on the fact that "different objectives may have conflicting gradient directions".

2. The term interaction profile is not very clearly defined mathematically. While the authors show design objectives within the interaction profiles section and including it in method figure 3, I find their connection and overall formulation very confusing.

3. The authors have not compared with any related work (if there exists any) but have only shown ablation comparisons. With task information guidance, I will automatically expect DGDM to work better than its unguided alternative without any task information. It is surprising to see that unguided DM is also solving tasks with considerably high success rates. For comparison with gradient descent, the initialization of the manipulator geometry might make some difference.

I would like to thank the authors for presenting most of the work very clearly.

**Quality Of The Limitations Section:**

3

**Questions For Rebuttal:**

1. Clarifying what exactly is an interaction profile. For example, a list of task objectives and their corresponding interaction plots will be very helpful.
2. Clarify figure 3 more. Particularly, is "current interaction profile" showing $\Delta\theta$ vs $\theta$ plot? How do you know $\theta$ as initial pose is unknown? What are "sampled poses"? How do you get "sampled poses" while reverse sampling?
3. Is it possible to compare with any possible related work? Also, if it is possible, initializing the start of optimization baseline with the output of unguided diffusion model might help in understanding the actual contribution of classifier guidance as compared to optimization.

**Robotics Focus:**

4

**Summary Of Paper:**

Diffusion model based designing of gripper fingers for a specified set of tasks and objects

**Summary Of Recommendation:**

With some more clarifications and justifications, the proposed work might be a good contribution to the community.

---

### Official Review · Reviewer_fWCT · 2024-07-20
**Review comments**

**Originality:** 4
**Technical Quality:** 4
**Clarity Of Presentation:** 5
**Potential Impact:** 3
**Recommendation:** 3
**Confidence:** 4

**Review:**

**Strengths**:
- The paper is very clearly written and was a joy to read.
- The work is well-motivated, and the contributions are placed within the context of existing work
- Mos design choices are justified and discussed in sufficient detail
- The proposed method is fast and effective (allowing for rapid prototyping)
- The learned model seems to be agnostic to the object shape
- The experiments include hardware demonstrations

**Weaknesses and questions**:
- The target orientation for the convergence task is a bit confusing to me. Is it specified by the user (I would guess so)? But if yes, why is it different for the proposed method and the unguided baseline in Fig. 5? Was the target orientation randomized in the experiments to test if the proposed approach can generalize in that dimension? Why is the convergence error to the target orientation not reported? Was it always zero for all methods?
- The reason for reporting the best of 16 trails (with varying initial conditions) instead of the statistics over all trails needs to be better discussed.
- It would be interesting to see the task progress over the 40 open-close actions. Do the baselines take more or fewer actions to converge compared to the proposed approach?
- Why was the success criterion defined as a threshold for the simple tasks, instead of a target?

**Quality Of The Limitations Section:**

3

**Questions For Rebuttal:**

Please address the questions and concerns raised above.

**Robotics Focus:**

4

**Summary Of Paper:**

The paper presents a data-driven framework for two-finger gripper design for sensor-free planar object reorientation via parallel closing action. The paper i) learns general (parallel-closing) interaction dynamics between arbitrary finger and object shapes, and ii) uses the learned dynamics model to guide a diffusion process that generates suitable gripper shape.

**Summary Of Recommendation:**

The paper proposes an efficient and effective method for two-finger gripper design that is agnostic to the object shape. Experiments are convincing to me and the paper is easy to follow. My only reservation centers around clarity on a couple of things that I have explained in my comments.

---

### Official Review · Reviewer_uUTj · 2024-07-21
**Interesting idea on using diffusion models and classifier guidance to generate task-specific finger designs for object manipulation.**

**Originality:** 4
**Technical Quality:** 4
**Clarity Of Presentation:** 4
**Potential Impact:** 3
**Recommendation:** 3
**Confidence:** 2

**Review:**

The paper is clearly written, precise and with enough detail in the main text to be well understood.

I particularly like the idea of using diffusion models as a prior for manipulator design and, as far as I know, this work is novel (although I’m not an expert in this field).

I appreciate that the authors build the paper by first showing the problems with the baseline (gradient descent) and only then introduce the necessity of using diffusion models and classifier guidance. All the properties of diffusion models - multimodality, handling data representations, gradient guidance - are well motivated, and experiments have been done to highlight these properties.

The experiment section shows convincing results that help understand the proposed method. The real-world experiments show that the method transfers well from simulation to reality.

One downside of this work is doing open-loop control. I’m not sure if introducing observations to the model would help in solving some of the failure cases (maybe the authors can comment on that), but I’d expect that for these types of interaction tasks, a closed-loop controller could work better.

**Quality Of The Limitations Section:**

3

**Questions For Rebuttal:**

Could you clarify how gradient descent is implemented? Are you doing some form of trajectory optimization using the dynamics network (which is a kind of dynamics model)? Are the gradients of the objective function “well behaved”, or would it be worth to try a gradient free optimiser (e.g. CMA-ES)?

It’s unclear to me if you have access to the initial object orientation. You state: “How the object should rotate depends on the initial orientation θ relative to the target orientation θ_target.” Are you running some pose estimation method? In that case, why not use it as a feedback signal and run an open loop? Wouldn’t it be more sensible for this kind of interaction task?

Could you clarify if, for each task description (e.g., shift down, shift right, …), you train a diffusion model (and dynamics network), or is there a single model conditioned on the task description?

**Robotics Focus:**

4

**Summary Of Paper:**

This paper introduces a novel approach to generate finger designs by leveraging diffusion models and a tailored guidance function. The authors propose to learn a dynamics network/model that uses as input an object representation and its pose (the state), along with the manipulator representation/parameters (the action), an outputs a delta in the objects pose (the next state).  Given a new task, e.g. ”shift the object down”, this model is first used to optimise a design objective/cost function F, by running gradient descent with respect to the manipulator parameters. Noting the problems of using gradient descent for optimizing F, namely convergence to local minima, which leads to very similar designs and task failure, the authors propose to learn an (unconditional?) diffusion model on the manipulator parameters, given data obtained in simulation.  Using this diffusion model and the previously learned dynamics network, the authors employ classifier-guidance to leverage multimodality, design precision and faster inference in comparison to the gradient descent baseline.

**Summary Of Recommendation:**

The paper idea is very interesting and the method and contributions are clearly explained.  The results show convincing results, in particular the sim2real transfer.

---

### Author Rebuttal · Authors · 2024-08-07

We thank all reviewers for their constructive suggestions and valuable comments, as well as for their appreciation of the novelty of the proposed idea and the promising results of manipulator design in both simulation and the real world. Below and in the rebuttal pdf, we address the main concerns and individual questions.

---

### Decision · Program_Chairs · 2024-09-04

**Decision:**

Accept

**Comment:**

**Post-Rebuttal Guidance**

Recommendation: ACCEPT.
Justification: Reviewers unanimously decided to accept. Rebuttal further mitigated any concerns or weaknesses the reviews had


**Pre-Rebuttal Summary**

Summary:
The authors present an approach to generate 2-fingered gripper designs for sensorless manipulation of a unknown part. The method involves a dynamics guided diffusion model, and shows promising results on both sim and real.


Strengths:
- Well written
- Well motivated
- proposed approach is novel, fast and generalizes to new shapes

Weaknesses:
- limited generalization to wider distribution of pose orientations
- lack of comparison to related work, only ablations shown
- interaction profile not well described, questionable justification/explanation for task guidance

Dear authors, looks like the reviewers have diverging opinions on the submission as well as several clarification questions. Hope you could address the reviewer concerns.